# Assessing the Financial Viability and Sustainability of Circular Business Models in the Wine Industry: A Comparative Analysis to Traditional Linear Business Model—Case of Georgia

Vakhtang Chkareuli [1],* , Gvantsa Darguashvili [2],*, Dzintra Atstaja [3] and Rozita Susniene [4]

1    Finance Program, Faculty of Business and Technology, Business and Technology University, 0162 Tbilisi, Georgia
2    Faculty of Business and Technology, Business and Technology University, 0162 Tbilisi, Georgia
3    Faculty of Law, Riga Stradins university, 1010 Riga, Latvia; dzintra.atstaja@rsu.lv
4    School of Economics and Business, Kaunas University of Technology, 44249 Kaunas, Lithuania; rozita.susniene@ktu.lt
*    Correspondence: vakhtang.chkareuli@btu.edu.ge (V.C.); gvantsa.darguashvili@btu.edu.ge (G.D.)

**Abstract:** In the contemporary global context, waste management and the judicious utilization of resources have emerged as pressing concerns. Consequently, the concept of a circular business model has gained prominence as a viable solution. This innovative model reframes waste not as a disposable byproduct but as an opportunity to generate new value, setting it apart from the conventional linear business model, particularly in financial, economic, and operational dimensions. Numerous industries grapple with the issue of excessive waste generation, among them the wine industry, notable for its substantial water and grape waste outputs. This predicament holds significant ramifications both on a global scale and within the specific context of Georgia. Yet, it also presents an innovative avenue for waste recycling. This study draws upon a comprehensive review of internationally recognized literature, noted for their scholarly significance and citation prevalence. In its practical segment, two distinct investment projects have been meticulously developed which seek to evaluate the financial viability of the circular business model in contrast to the conventional linear business model. The investment projects considered are as follows: 1. Under the framework of a linear business model, the company exclusively engages in the production and sale of wine. 2. Within the circular business model paradigm, the company not only produces wine but also harnesses waste processing to yield grape seed oil, which is subsequently marketed alongside wine bottles. Both models undergo rigorous scrutiny, employing a comprehensive analysis of key financial indicators essential for assessing project profitability and efficiency. The outcomes of this investigation reveal that, under identical capital investment conditions, the circular business model surpasses the linear model in terms of profitability. This underscores the potential for sustainable practices within the wine industry and the broader business landscape.

**Keywords:** circular business model; linear business model; wine industry; sustainability; financial analysis

## 1. Introduction

The contemporary challenge of waste management, exacerbated by global environmental degradation, highlights the inadequacies of the linear economic model, a paradigm focused on a "take, make, dispose" approach that leads to unsustainable depletion of natural resources. Since the 1970s, this critical issue has been at the forefront, giving rise to the circular economy as a more sustainable economic paradigm [1]. The circular economy, integrating economic principles, management strategies, financial mechanisms, and technological innovations, redefines waste as a valuable resource for future production, thus breaking away from the linear model's limitations [2].

In today's era, where sustainable development is paramount, the efficient use of Earth's finite resources and the preservation of the biosphere are of utmost importance [3]. Transitioning from entrenched linear business models to circular approaches presents significant challenges, especially in terms of initial investments and liquidity [4]. However, this transformation, essential for long-term environmental and financial sustainability, offers considerable benefits. Studies by Geng et al. [5], Sarkis et al. [6], Geissdoerfer et al. [7], Weetman et al. [8], Dzwigol et al. [9] and Aureli et al. [10] have explored the implications of this shift in various contexts, highlighting its potential for enhancing profitability through efficient waste management and resource use. In the Georgian wine industry, this transition's financial viability and sustainability are particularly crucial, as the sector stands at the crossroads of traditional practices and innovative, sustainable approaches.

In their study, Mura et al. [11] investigate the adoption of circular economy principles within the wine industry, focusing on sustainable packaging initiatives. The research underscores the crucial influence of environmental policies in facilitating this transition, identifying both obstacles and drivers for the integration of circular packaging methods. The study underscores the strategic significance of policy frameworks in supporting the wine sector's movement towards eco-friendly practices. By demonstrating the role of policy-driven strategies in promoting circular business models, the research offers insights into the potential environmental and economic benefits for the wine industry.

This research aims to investigate the economic implications of adopting circular business models in the wine industry, focusing on their potential to improve sustainability metrics without compromising financial performance.

*Research Questions*

1.  How does the financial viability of circular business models in the Georgian wine industry compare to traditional linear models?
2.  What are the financial implications of adopting circular business models for sustainability in the wine industry?

    Hypotheses:

3.  The circular business model in the Georgian wine industry demonstrates greater financial viability than the traditional linear model due to enhanced resource efficiency and value creation from waste.
4.  Implementing circular business models in the wine industry leads to significant improvements in sustainability metrics without compromising financial performance.

These questions and hypotheses are designed to guide the investigation of the financial and sustainable aspects of different business models within the context of the Georgian wine industry.

The primary objective of this research is to conduct a practical, comparative, financial analysis of circular and linear business models. This objective is achieved through two primary tasks and several sub-tasks:

1.  Examination and study of the transformation of traditional business models in various wine companies globally and within Georgia towards circularity.

    -   Investigation of practices implemented in the international and local wine market within a circular business model.
    -   Deliberation on critical aspects of transformative changes.
    -   Description of the practical processes distinguishing the circular business model from the linear business model.

2.  Development of a comprehensive investment project and the construction of a long-term financial model, illustrating the transformation from a linear to a circular business model, using a wine company as an illustrative example.

    -   Preparation of the investment project based on market-appropriate assumptions and anticipated outcomes.
    -   Composition of the financial model encompassing profit and loss statements, financial position analyses, and cash flow projections.

- Recalculation of key coefficents based on the investment project and financial model, leading to the formulation of conclusions and recommendations.

The significance of this research lies in its examination of the financial perspective of the Georgian wine market and its readiness to embrace a circular business model. It underscores the potential for sustainable practices and circular economy principles to mitigate waste-related challenges within the industry, which later can be generalized through the global wine market.

The circular business model, as previously discussed, represents a contemporary approach aimed at generating added value from production waste and optimizing resource utilization efficiency. In its essence, the circular economy provides a fundamental framework for fostering sustainability.

## 2. Literature Review

This literature review adopts a systematic approach, methodically evaluating and synthesizing existing research on circular and linear business models in the wine industry. It aims to identify, analyze, and summarize the key findings related to the financial performance and sustainability of these models, with a specific focus on the context of Georgia. The review seeks to provide a thorough understanding of the current state of knowledge in this area, highlighting gaps and opportunities for future research.

The selection of literature for this research was inspired to explore the financial viability and sustainability of circular business models in the wine industry, as compared to traditional linear models. Focused on the specific context of Georgia's wine industry, the criteria encompassed studies on the operational, economic, and environmental impacts of both circular and linear models. Emphasis was placed on literature that addresses waste management, resource efficiency, and sustainable practices within agricultural sector as a whole and in viticulture and oenology in particular, including technological innovations and policy frameworks influencing the industry. Additionally, financial analyses and sustainability metrics pertinent to the wine industry's business models were crucial in this selection, aiming to provide a well-rounded understanding of the sector's challenges and opportunities for transitioning towards more sustainable practices.

The critical challenge of waste management in the contemporary world, exacerbated by global climate degradation and environmental pollution, calls for a transformative shift from the linear "take, make, dispose" economic model to a more sustainable circular economy [1]. This shift is underscored by recent research, which highlights the circular economy's potential to mitigate negative impacts and foster financial viability through efficient waste management and resource use [5–7].

Kanda et al. [12] explore the evolution from circular business models to ecosystems, emphasizing the interconnectedness of various stakeholders in achieving circularity. This concept extends to the wine industry, where collaboration across the value chain can enhance sustainability. The studies by Aureli et al. [10] and Dzwigol et al. [9] delve into the role of management and environmental accounting in the circular economy, highlighting the importance of tracking and optimizing resource flows.

The research by Batova et al. [13] focuses on the challenges and opportunities in adopting circular practices, noting the importance of overcoming cultural, regulatory, and technological barriers. This is particularly relevant for industries undergoing rapid transformation, like the Georgian wine industry. Further, the work by Weetman et al. [8] provides insights into the financial implications of this transition, emphasizing the potential long-term economic benefits despite initial capital costs.

The organizational and economic mechanisms for implementing green logistics, as discussed by Dzwigol et al. [9], are crucial for industries looking to integrate circular principles. This approach could be instrumental in redefining logistics and supply chain practices in the wine industry, leading to more sustainable operations.

Cavicchi & Vagnoni [14] explore the integration of Energy Management (EM) within circular economy models in the agricultural sector, specifically in high-tech hydroponic

tomato cultivation. This study reveals how EM capabilities are crucial for developing organizational competencies that facilitate the adoption of circular economy principles, enhancing the firm's sustainability performance. By focusing on the agricultural SME's experience in Italy, the research provides a practical example of how energy management contributes to circular business models, highlighting the strategic role of EM in achieving sustainability goals. This insight is particularly relevant to the broader discourse on sustainable practices within industries like the Georgian wine industry, where the transition to circular models represents both a challenge and an opportunity for enhancing environmental and financial sustainability.

*2.1. Wine Production and Environmental Impact*

Over recent decades, the global popularity of food and wine tourism has surged, giving rise to significant developments within these industries. Alongside this growth, the imperative of environmental sustainability and protection has become increasingly salient.

The 2021 statistical report on global viticulture reveals a wine production volume of 260 million hectoliters worldwide, with 60% originating from the European region. Notably, the European region is renowned for its thriving wine industry, but it concurrently grapples with a surplus of waste. On average, the production of 1 ton of grapes yields 750 L of wine, along with 1650 L of wastewater and approximately 250 kg of solid waste [15] These figures underscore the substantial water consumption associated with wine production, with an estimated 1–4 L of water required for every liter of wine produced [16].

Applying the data from 2021, it becomes evident that the wine production process utilized a staggering 260 to 1040 million hectoliters of water in that year. Regrettably, much of the waste generated in this industry remains underutilized.

The wine production process unfolds through several pivotal stages. Initially, grapes are harvested, destemmed, and partially crushed. Subsequently, they are transferred to an open container, known as a fermenter, where the process of sugar fermentation commences. After fermentation, the juice is separated from the grape skin, seeds, and residual stems (resulting from grape pressing). The clarified juice is then moved into closed vessels for aging. Meanwhile, the remaining grape solids and sediments settle at the bottom of the container. Once the wine has matured sufficiently, it is bottled [16].

The potential for processing solid waste generated during wine production is notably substantial, and its application holds promise across various sectors, including agriculture, medicine, and energy. The surplus water remaining after production could serve as a primary source for irrigation canals following suitable treatment and renovation. This transition has the potential to significantly curtail the demand for fresh water resources. Consequently, the implementation of a circular business model within the wine industry presents a tangible opportunity to derive financial value from excess waste generated during the production process, thereby generating income and profit [17].

The United Nations Tourism Organization currently places paramount importance on the harmonious coexistence of tourism and the environment. It insists that all forms of tourism should be equipped to preserve the pristine state of the biosphere [18]. As Montella elucidates in the article titled "Wine Tourism and Sustainability: An Overview", numerous countries have prioritized sustainable development, leading to a considerable portion of the global population gravitating towards green tourism.

The wine industry, as one of the leading sectors, has embarked on optimizing resource utilization and safeguarding the biosphere. Despite the prominence of green tourism as a contemporary global theme, certain countries still exhibit limited awareness regarding the sustainable development of the biosphere. This gap in understanding inevitably reverberates through various industries, including the wine sector.

Consequently, the successful transformation towards a circular business model necessitates the diligent development of new technologies and processes, naturally precipitating a restructuring and refinement of the production chain.

### 2.2. Types of Waste in Wine Production and Its Challenges

The wine industry is characterized by an excessive volume of waste, notably rich in biodegradable components. Defining sustainability within this industry can be intricate, with potential outcomes ranging from cost reduction to the generation of novel products [19].

Throughout the various stages of wine production, a diverse array of waste materials is generated, encompassing both solid and liquid forms:

1.  Grape Leaves: These are the residual remnants from the grape harvest, a relatively underexplored waste stream. Available information on their composition suggests that they contain a diverse array of compounds, including acids, enzymes, vitamins, tannins, and sugars, which make them potentially valuable for a circular business model.
2.  Grape Stalks: Following the removal of grape stems, residual stalks remain, constituting approximately 1.4% to 7% of the processed raw material. While grape stalks have limited commercial value, they are sometimes utilized in soil nutrient formulations.
3.  Solid Grape Residues from Pressing: During grape juice production, various types of waste are generated, offering multifaceted potential applications. For example, grape seeds can be processed into oil, suitable for use in pharmaceuticals, cosmetics, and the food industry as a natural source of antioxidants.
4.  Wine Sediment: This waste product accumulates at the bottom of wine barrels during the winemaking process. It contains highly bioactive molecules that can be harnessed to produce extracts or semi-finished products for the food and pharmaceutical industries.
5.  Wine Wastewater: In contemporary society, the responsible management of agricultural soil and water resources is of paramount concern. On average, every ton of grapes yields 3000 to 4000 L of wastewater, containing various organic and inorganic pollutants that pose a significant environmental burden. Properly treated wine wastewater has the potential to be repurposed for farm irrigation, thus reducing environmental impact and minimizing irrigation costs.

Given the rapid growth of the wine industry, particularly in Europe, there is a corresponding increase in its environmental footprint. Effective management of solid and liquid waste from wine production remains a major challenge for the industry [16]. Despite the significant volume of waste generated, which could serve as a foundation for creating new circular economy products, technological readiness within the wine industry, as well as in various other sectors, remains relatively low.

Italy, a prominent player in the global wine industry, actively embraces the principles of the bioeconomy. This entails a transition from a conventional, fossil fuel-dependent economy to a more sustainable and environmentally friendly model through modern technologies, renewable resource utilization, and organic product production, all aimed at minimizing environmental impact. Numerous studies explore the potential of implementing a circular economy in wine production, focusing on the recovery of biologically valuable products from wine production waste, such as grape oil [20].

Notably, the linear business model still predominates globally, underscoring the primary aim of this article: to assess the ecological sustainability of wine production, exemplified by Italy, from two perspectives—the linear and circular business models.

### 2.3. Transformation to a Circular Business Model

The production chain within the linear wine business model comprises three distinct phases:

1.  Agricultural Phase: This encompasses the entire spectrum of processes necessary for vineyard management and grape harvesting, including soil treatment, pesticide application, and harvesting.
2.  Wine Production Phase: In this stage, grapes are transformed into the final product, wine. It includes grape sorting, pressing, fermentation, wine aging, and maturation.
3.  Bottling Phase: Here, wine is bottled in glass containers, typically sealed with corks, capsules, and labels.

However, we have to explore the transformation of these three primary production phases under a new circular business model, examining various scenarios:

1.  Enhanced Agricultural Phase: This scenario considers the replacement of conventional fuels with biofuels, thereby reducing environmental impact. Additionally, the production of pomace oil from grape waste is explored, with applications spanning various industries. Furthermore, the introduction of bio-fertilizers could potentially result in a 50% reduction in fertilizer use.

2.  Enhanced Wine Production Phase: This stage explores avenues to reduce electricity consumption, such as utilizing steam derived from vineyard pruning.

3.  Enhanced Production Chain: Replacing industrial steam with biologically derived steam obtained from pruning stalks and stems is investigated. This scenario also considers the production of calcium tartrate, a versatile compound with wide-ranging applications in the food, pharmaceutical, and wine industries.

Wastewater stemming from wine production is characterized by an acidic pH and an excess of harmful organic substances, posing potential threats to the hydrosphere and its inhabitants. Research highlighted in the article suggests that the implementation of complex chemical and biological wastewater treatment processes can yield clean water resources, offering a novel opportunity for irrigation canal replenishment. Solid residues, including grape pomace and skin, are identified as rich sources of carbohydrates and lipids, showcasing significant potential for utilization in food products, cosmetics, pharmaceuticals, and agricultural fertilizers. Additionally, grape stems are recognized for their potential to enhance the flavor intensity, purity, color, and shelf life of bottled wine. The cellulose and hemicellulose present in these solid grape residues hold the potential for conversion into biofuel, representing a financially attractive and environmentally beneficial energy source [20].

In the context of evolving industries, the concept of a biofactory, capable of generating valuable products through waste processing while minimizing environmental harm, emerges as a harbinger of the future. Although residues from the wine industry have been identified as promising feedstocks for integrated biorefinery applications, a dearth of practical studies on comprehensive recycling processes persists. Crucially, the transformation of key elements such as fuel, fertilizers, pesticides, electricity, water consumption, and glass bottles stands as a pivotal endeavor in aligning wine production with the circular business model. This transformation ensures that environmental considerations are integrated into the production process for the betterment of the biosphere. The research encapsulated within this article convincingly underscores the advantages attainable through the adoption of a novel circular business model within wine production. Recycling existing waste products, such as grape seed oil, calcium tartrate, and others, is poised to contribute significantly to both economic and environmental aspects of the industry [20].

### 2.4. Wine Industry in Georgia

Agriculture has long thrived in Georgia, supported by its advantageous geographical location, fertile soil, and favorable climate. Viticulture, in particular, boasts a rich history dating back 8000 years and continues to hold a pivotal role in the country's economic landscape. Consequently, our research focus centers on the wine industry, a domain that garners both global and local significance, offering a diverse and compelling subject of inquiry.

Georgian wine production stands as a testament to an illustrious heritage, dating back 8000 years, rivaling renowned wine-producing nations such as France, Italy, and Spain. In recent years, Georgia has gained increasing recognition as the "cradle of wine", earning accolades at international wine competitions. Notwithstanding its growing prominence and advancements in wine bottling technologies that blend modernity with traditional practices, the global populace demonstrates a growing preference for organic wines. Furthermore, there is a burgeoning interest in companies that recognize the value inherent in production waste—an aspect still in the developmental stages in Georgia.

## 2.5. Global Wine Market Trends and Sustainability

The literature review incorporates scientific and research insights highlighting the wine industry's preeminence within the global agricultural sector. This distinction is not only rooted in substantial income generation but also in the generation of significant waste, aligning with the principles of the circular economy, wherein waste represents a tangible opportunity for value creation. Moreover, we emphasize that the transition to a circular economy is now being adopted by a majority of developed countries and select developing nations.

The practical facet of our research centers on the establishment of a wine company within the Georgian market, accompanied by the evaluation of an investment project and its corresponding financial metrics. This segment comprises three essential components:

1.  Comprehensive Market Overview: This section provides a holistic view of the global and Georgian wine markets, shedding light on industry sales and consumption trends. It underscores the dynamism of the wine sector, which has evolved over millennia to adapt to new market dynamics and emerging trends.

2.  Financial Model Development: The financial model of the wine company is constructed using two distinct approaches. The first approach adheres to the traditional linear model, where the company solely generates income through wine sales. In contrast, the second approach embodies a circular and innovative model, wherein the company not only produces wine but also collects grape waste from local producers, repurposing it into grape seed oil.

3.  Comparative Financial Analysis: Both financial models are created with identical capital investments, facilitating a direct comparison of key financial indicators. This final phase involves the analysis and interpretation of results.

General insights into the wine market reveal its expansiveness, diversity, and ongoing growth. Wine production, with roots extending thousands of years, has deeply embedded itself within the cultural, social, and economic fabric of various regions. The industry is remarkably dynamic, continually evolving to adapt to new market dynamics and trends.

As the market matures, the wine sector has increasingly gravitated toward environmentally responsible production and sustainability. Wineries in many leading countries have transitioned to organic, biodynamic, and sustainable farming practices to mitigate the adverse environmental effects of inorganic chemicals, minimize waste disposal, and maximize resource utilization.

Over recent decades, the global wine industry has witnessed remarkable growth in sales and profits, with wine consumption on the rise in numerous countries, including developing nations. In 2021, global wine consumption reached 236 million hectoliters, with Georgia producing 107 million bottles of wine putting Georgia 18th among wine-exporting countries, with a 0.6% market share of the total [21].

The escalating growth rate of the wine market correlates with a commensurate increase in its environmental footprint. This phenomenon arises from the imperative faced by wine companies to meet rising consumer demand, resulting in heightened production volumes and, consequently, greater waste generation. Regrettably, a substantial portion of this waste continues to find its way into landfills. It is noteworthy, however, that the global wine industry has undertaken concerted efforts in recent years to champion environmental conservation, thereby garnering favor from consumers, businesses, and the ecosystem. Consequently, a diverse and intriguing array of responses to this issue has emerged among wine companies.

Evolving patterns in wine consumption underscore a shift toward quality over quantity, with consumers displaying a predilection for organic wines. Furthermore, heightened attention is being devoted to wine packaging, labels, and design aesthetics. Biodegradable packaging materials, in particular, have garnered favor among consumers. Reflecting prevailing trends, sustainability certification has assumed paramount importance in brand promotion. Wines boasting USDA certification are characterized by the absence of fertilizers, pesticides, and various chemicals in their production processes. Even in the absence of formal certification, wineries are increasingly committed to minimizing their environmental

footprint. This entails adopting renewable energy sources, conserving water resources, implementing waste recycling initiatives, and preserving the integrity of fertile soils.

Illustratively, a recent and noteworthy sustainable practice has emerged within the wine market—an innovation centered on the utilization of paper wine bottles. This pioneering approach substantially mitigates the carbon footprint, reducing environmental impact by an impressive 84%. The British company "Frugalpac" has successfully pioneered the production and implementation of such bottles, setting an influential precedent for other industry players to adopt sustainable packaging methods.

Numerous enterprises operating within the wine industry have embarked on transformative initiatives involving the repurposing and commercialization of waste materials. For instance, the Australian personal care company, "Swisse", has actively explored the utilization of vineyard waste in the creation of nutritional supplements and skincare products [22]. Similarly, within the Georgian market, the Shukhman Group, an entity owning vineyards and engaged in wine production, initiated the production and sale of grape seed oil scrub in 2019—an innovative venture within the local market.

As previously noted, the United States of America stands as the most lucrative market for wine consumption. Notably, two of its regions, Napa and Sonoma Valley, are renowned for their exceptional wines. However, these regions have faced daunting challenges in recent decades, characterized by rampant forest fires and severe droughts. In response, vineyards have returned to the practice of "dry farming", a method that entails maintaining soil moisture through non-irrigation means. This practice forces the vine to adapt to arid conditions, with winemakers frequently asserting that wines produced via "dry farming" techniques exhibit superior taste. Beyond the United States, the Greek islands serve as another compelling exemplar of "dry farming." These islands, despite minimal precipitation—sometimes as little as a few centimeters annually—yield globally recognized and delectable wines. In an era marked by the increasing scarcity of water resources, practices like "dry farming" are poised to gain prominence within the viticultural landscape [22].

Moreover, the eco-friendly grape seed oil, derived from grape seeds post-crushing during wine production, has witnessed a surge in popularity. France pioneered large-scale production of this product, and today, numerous enterprises worldwide engage in its manufacturing [23].

In 2021, the international grape seed oil market reached a substantial valuation of 473 million dollars. Anticipations suggest a further surge in demand for this product in the forthcoming years. This projection is grounded in the multifaceted utility of grape seed oil, known for its anti-inflammatory, acne-reducing, and moisture-retaining properties. Additionally, grape seed oil finds applications in massage therapy, alleviating muscle pain, and addressing cardiovascular ailments.

The contemporary era witnesses a significant portion of the global populace embracing a health-conscious lifestyle, demonstrating a heightened regard for personal and collective well-being. Consequently, these individuals represent a burgeoning consumer base for grape seed oil, owing to its evident health benefits.

As earlier underscored, Georgia's prominence and recognition within the global wine industry continue to ascend annually, underscored by an array of international accolades. The nation's foremost objective has become the production and dissemination of high-quality wine products. This approach aims to minimize the risk of reputational threats while ensuring that Georgia's captivating history, culture, and traditions remain firmly in the global spotlight.

## 3. Results

The results of this study, focusing on the comparative financial analysis of circular and linear business models within the wine industry, demonstrate the potential for generalization across diverse companies. However, it is critical to recognize that the observed profitability and financial metrics are not solely dependent on the choice of business model.

They are also significantly influenced by a range of financial variables, including the cost of goods sold (COGs), variable and fixed costs, and the costs associated with debt and equity etc. This underscores the complexity of financial performance in business models and the necessity of a nuanced understanding of how these variables interact within the unique context of each company.

### 3.1. Methodology

This research employs a mixed-method approach to analyze the financial viability and sustainability of circular and linear business models in the Georgian wine industry. It interweaves quantitative financial analysis with qualitative assessments to provide a holistic view of these business models.

Development of Investment Projects and Data Collection: Central to our methodology is the development of two distinct investment projects that model the financial implications of both circular and linear business models within the Georgian wine industry. We formulated specific assumptions, gathered targeted data, and conducted relevant calculations, tailoring them to the unique aspects of this industry.

For quantitative data, we sourced financial statements for the year 2021 from select wine companies operating in the Georgian market. These companies, owning vineyards of up to 200 hectares primarily in the Kakheti region and deriving their main revenue from wine sales, served as our primary data sources. This data collection was supplemented by international desk research, where we derived quantitative assumptions about grape yields, wine production, production waste, and grape seed oil.

Qualitative Research and Literature Review: The qualitative component of our research hinges on an exhaustive literature review, encompassing studies on circular and linear economic models both globally and within the Georgian wine industry. This review informed our understanding of the sustainability practices and financial structures of these models.

Financial Analysis: Our financial analysis involved a detailed examination of key financial indicators, such as profitability and efficiency, under both business models. This involved comparing the financial performance of the circular model against the traditional linear model, using real-world data and the developed investment projects as bases. Sustainability and Historical Context: The research also delves into the sustainable development and waste recycling aspects of these business models, with a particular focus on the agricultural sector. Georgia's rich heritage in grape cultivation, which spans millennia, and the enduring popularity of viticulture, make the wine industry an ideal case study for exploring the adaptation to circular business models.

Ethical Considerations and Limitations: Ethical considerations include ensuring the confidentiality of the financial data obtained. The study acknowledges potential limitations, such as the regional focus on Georgia and the specificities of the selected companies.

This comprehensive methodology, combining investment project development, financial analysis, and a relevant literature review, is designed to provide an in-depth understanding of the financial and sustainability implications of circular and linear business models in the Georgian wine industry.

### 3.2. Financial Models for Wine Companies: Evaluating Linear and Circular Business Models

Recent data unequivocally indicate an upward trajectory in future sales within the Georgian wine market. This growth is expected to be directly proportional to the increase in wine production waste. In light of the imperative to sustain environmental resources, it has become essential to embrace, cultivate, and implement modern technologies within the wine production process. These technologies will augment the capabilities of wine companies by facilitating waste utilization, recycling, and the creation of value-added products.

As previously mentioned, the global wine industry is actively engaged in the pursuit of sustainable development. Numerous international companies are fervently embracing and advancing diverse methodologies to conserve environmental resources, employ organic

fertilizers, and curtail the disposal of waste in landfills [24]. Nevertheless, as elaborated in the initial segment of the market research, industrial waste processing technology in Georgia is still in its nascent stages and has yet to attain adequate development within the agricultural sector. Consequently, only a select few wine companies have adopted contemporary approaches, manufacturing products like grape seeds and seed oil from industrial waste while also incorporating natural, organic fertilizers into their practices.

Given the rapid pace of technological advancement on a global scale, we contend that it is imperative for all leading industries, including the wine sector, to exert maximal efforts and implement transformative changes. To facilitate a comprehensive examination of the research topic, this study presents two investment projects for a wine enterprise operating within the Georgian market: one employing a linear business model and the other adopting a circular business model. Subsequently, it undertakes a comparative analysis of the results and conducts a financial assessment spanning a decade. It is pivotal to note that the principal assumptions remained consistent throughout the development of both models:

1. A uniform investment amount of 10 million GEL is allocated to the company in both scenarios, enabling a direct comparison of the relative profitability of each model with identical initial capital.
2. The vineyard is established in the Kakheti region, widely regarded as the cradle of winemaking. The total vineyard area spans 200 hectares.
3. The cultivation exclusively focuses on the Rkatsiteli grape variety, chosen for two primary reasons: Rkatsiteli ranks among the most prevalent grape varieties in Georgia, particularly within the Kakheti region. Furthermore, when waste results from a single grape variety, the production of grape seed oil is optimized.
4. Regarding vineyard yields, each hectare yields 5000 kg of grapes, ultimately producing 3750 L of wine. Based on these figures, a total harvest of 1000 tons is projected for the 200-hectare vineyard, yielding 750,000 L of wine.
5. It is assumed that all company products are sold within the same fiscal year of production.
6. Sales are projected to experience an annual growth rate of +10%, commencing from the third year onwards. This percentage is chosen based on the observed 13.8% increase in Georgian wine exports in 2021 compared to 2020 [22].
7. A 2% annual increment in sales prices is anticipated. Historical data indicates fluctuations in pricing trends over the last five years; therefore, a conservative approach is adopted, forecasting a modest positive trend.
8. Capital expenditures are incurred in the initial year to facilitate vineyard acquisition, factory construction, and machinery procurement. Consequently, no sales revenue is expected in the first year [25].
9. The company employs a workforce of 100 individuals, comprising 30 administrative staff and 70 factory workers. This staffing assumption mirrors the employment structure of a Georgian wine company managing up to 200 hectares of vineyards in Kakheti.
10. A discount rate of 12% is selected as the optimal rate within the business environment, considering that the refinancing rate fluctuates between 10–11%.
11. Three forms of financial reporting are provided, encompassing profit and loss statements, financial position statements, and cash flow statements.

3.2.1. Linear Business Model

Within the framework of the linear business model, the operational process is straightforward and conventional: the company engages in wine production and disposes of production waste, deriving its revenue solely from the sale of wine bottles.

Profit and Loss Statement, shown in Figure 1:

1. Revenue: Investment activities commence in the first year, as noted earlier, rendering sales revenue nonexistent in this initial year. Subsequently, sales are initiated in the second year of operation. The projected yield, based on the 200-hectare vineyard, amounts to 1,000,000 wine bottles (standard size of 0.75 L each). According to researched data, the selling price per bottle is $2.5 at the prevailing exchange rate of

6.30 Lari. Consequently, the sale of wine generates an income of 5.4 million GEL for the company In the second year. In subsequent years, a 10% increase in production quantity and a 2% rise in the sales price are factored in annually, leading to corresponding increments in income [26].

2. Cost of Goods Sold (COGs): The cost price per wine bottle encompasses expenses for 1 kg of Rkatshiteli grapes, packaging materials, and the salary of a factory employee directly involved in the production process. The grape cost aligns with the market price, with an annual escalation of 2% in tandem with the rise in sales price. The bottle, labels, thermocache, and stopper costs are determined based on market rates [27], with an annual projected increase of 2%, in harmony with the sales price hike. Regarding salaries, as 70% of the company's workforce operates within the factory, 70% of the total salary expense is allocated to the cost component, while the remaining 30% pertains to operating expenses.

3. Operating Costs: Salary expenses are calculated based on the average monthly salary data from 2018–2022, averaging at 1176 GEL [28], and then multiplied by the employee count. An annual 5% increase is incorporated. Marketing, transportation, consulting, insurance, and fuel expenses are recalculated as a percentage of sales revenue, with a range of 1–2%. Annual utility costs are determined according to local tariffs and the anticipated production consumption, accounting for water and electricity usage. Research indicates that irrigating 1 hectare necessitates an average of 2 million liters of water, while 4.74 L of water are required to produce 1 L of wine. Additionally, the bottling process consumes 22,540 kWh for every 1 million bottles, adjusted and calculated based on projected sales for each year.

4. Depreciation: Depreciation is calculated using the straight-line method, with the value of long-term assets being depreciated over an estimated useful life of 50 years.

5. Taxes: Taxes comprise property tax, equivalent to 1% of the value of fixed assets.

6. Contingency Expenses: Given that the presented financial model relies on assumptions, unanticipated costs may arise. Accordingly, a forecast allowance of 3% of sales is accounted for.

7. Net Income: Income and expenses are presented net of taxes.

This delineation outlines the financial aspects of the linear business model for the wine company.

| Profit and Loss Statement *All amounts are in EUR* | Year 1 | Year 2 | Year 3 | Year 4 | Year 5 | Year 6 | Year 7 | Year 8 | Year 9 | Year 10 |
|---|---|---|---|---|---|---|---|---|---|---|
| Revenue | 0 | 1,898,007 | 2,150,859 | 2,389,605 | 2,654,851 | 2,949,539 | 3,276,938 | 3,640,678 | 4,044,794 | 4,493,766 |
| COGs | 0 | -1,005,149 | -1,101,017 | -1,207,243 | -1,325,024 | -1,455,698 | -1,600,766 | -1,761,906 | -1,940,997 | -2,140,144 |
| Gross Profit | 0 | 892,858 | 1,049,842 | 1,182,362 | 1,329,827 | 1,493,841 | 1,676,172 | 1,878,772 | 2,103,797 | 2,353,622 |
| GP, % | 0% | 47% | 49% | 49% | 50% | 51% | 51% | 52% | 52% | 52% |
| | | | | | | | | | | |
| Salaries and other employe | 0 - | 159,287 - | 167,252 - | 175,614 - | 184,395 - | 193,615 - | 203,295 - | 213,460 - | 224,133 - | 235,340 |
| Sales and marketing expen | 0 - | 37,960 - | 43,017 - | 47,792 - | 53,097 - | 58,990 - | 65,539 - | 72,814 - | 80,896 - | 89,875 |
| Transportation expenditure | 0 - | 18,980 - | 21,509 - | 23,896 - | 26,549 - | 29,495 - | 32,769 - | 36,407 - | 40,448 - | 44,938 |
| Consulting expenses | 0 - | 18,980 - | 21,509 - | 23,896 - | 26,549 - | 29,495 - | 32,769 - | 36,407 - | 40,448 - | 44,938 |
| Insurance | 0 - | 16,241 - | 18,404 - | 20,447 - | 22,717 - | 25,238 - | 28,040 - | 31,152 - | 34,610 - | 38,451 |
| Fuel | 0 - | 13,249 - | 15,014 - | 16,680 - | 18,532 - | 20,589 - | 22,874 - | 25,413 - | 28,234 - | 31,368 |
| Utilities | 0 - | 88,964 - | 98,427 - | 106,737 - | 116,977 - | 128,242 - | 140,633 - | 154,263 - | 169,256 - | 185,749 |
| Depreciation and Amortiza | 0 - | 41,217 - | 41,217 - | 41,217 - | 41,217 - | 41,217 - | 42,217 - | 41,217 - | 41,217 - | 42,217 |
| Taxes | 0 - | 27,239 - | 26,827 - | 26,414 - | 26,002 - | 25,590 - | 25,178 - | 24,766 - | 24,354 - | 23,941 |
| Other OPEX | 0 | -6298 | -6740 | -7195 | -7684 | -8213 | -8783 | -9399 | -10065 | -10786 |
| Operating Revenue | 0 | 464,443 | 590,927 | 692,473 | 806,109 | 933,156 | 1,074,074 | 1,233,475 | 1,410,135 | 1,607,018 |
| | | | | | | | | | | |
| Profit tax | 0 | -4644 | -5909 | -6925 | -8061 | -9332 | -10751 | -12335 | -14101 | -16070 |
| Net Income | 0 | 459,799 | 585,018 | 685,548 | 798,048 | 923,825 | 1,064,324 | 1,221,140 | 1,396,034 | 1,590,948 |
| NI, % | 0% | 24% | 27% | 29% | 30% | 31% | 32% | 34% | 35% | 35% |

**Figure 1.** Profit and loss statement with linear business model. Source: Authors' calculations, Reporting Portal of Georgia [29], National Statistics Office of Georgia [28].

Statement of Financial Position, shown in Figure 2:

| Statement of Financial Position | Year 1 | Year 2 | Year 3 | Year 4 | Year 5 | Year 6 | Year 7 | Year 8 | Year 9 | Year 10 |
|---|---|---|---|---|---|---|---|---|---|---|
| *All amounts are in EUR* | | | | | | | | | | |
| **Assets** | | | | | | | | | | |
| Fixed Assets | 2,342,560 | 2,342,560 | 2,342,560 | 2,342,560 | 2,342,560 | 2,342,560 | 2,342,560 | 2,342,560 | 2,342,560 | 2,342,560 |
| Biological assets | 422,535 | 422,535 | 422,535 | 422,535 | 422,535 | 422,535 | 422,535 | 422,535 | 422,535 | 422,535 |
| *Accumulated Depreciation* | - | *(41,217)* | *(82,435)* | *(123,652)* | *(164,870)* | *(206,087)* | *(247,304)* | *(288,522)* | *(329,739)* | *(370,957)* |
| Inventory | - | - | 21,631 | 70,777 | 154,525 | 281,389 | 461,562 | 707,225 | 1,032,892 | 1,455,836 |
| Accounts Receivable | - | 462,929 | 524,600 | 582,830 | 647,525 | 719,400 | 799,253 | 887,970 | 986,535 | 1,096,040 |
| Prepaid TAX | - | 183,106 | 205,108 | 229,197 | 256,143 | 286,286 | 320,009 | 357,736 | 399,946 | 447,173 |
| Cash and Cash Equivalents | 756,032 | 1,081,237 | 1,664,534 | 2,318,873 | 3,048,321 | 3,857,307 | 4,749,982 | 5,730,063 | 6,800,641 | 7,963,940 |
| **Total Assets** | **3,521,127** | **4,451,150** | **5,098,534** | **5,843,120** | **6,706,739** | **7,703,390** | **8,848,597** | **10,159,568** | **11,655,370** | **13,357,122** |
| | - | - | - | - | - | - | - | - | - | - |
| **Shareholder Equity** | - | - | - | - | - | - | - | - | - | - |
| Share capital | 3,521,127 | 3,521,127 | 3,521,127 | 3,521,127 | 3,521,127 | 3,521,127 | 3,521,127 | 3,521,127 | 3,521,127 | 3,521,127 |
| Current Period Profit (loss) | - | 459,799 | 585,018 | 685,548 | 798,048 | 923,825 | 1,064,324 | 1,221,140 | 1,396,034 | 1,590,948 |
| Accumulated Profit (loss) | - | - | 459,799 | 1,044,816 | 1,730,364 | 2,528,412 | 3,452,237 | 4,516,560 | 5,737,700 | 7,133,734 |
| **Total Equity** | **3,521,127** | **3,980,926** | **4,565,943** | **5,251,491** | **6,049,539** | **6,973,363** | **8,037,687** | **9,258,827** | **10,654,861** | **12,245,809** |
| | | | | | | | | | | |
| **Liabilities** | - | - | - | - | - | - | - | - | - | - |
| Accounts Payable | - | 48,944 | 54,541 | 60,157 | 66,367 | 73,235 | 80,831 | 89,233 | 98,526 | 108,806 |
| TAX Payables | - | 421,280 | 478,049 | 531,472 | 590,833 | 656,791 | 730,079 | 811,508 | 901,983 | 1,002,507 |
| **Total Liabilities** | **-** | **470,224** | **532,591** | **591,629** | **657,200** | **730,026** | **810,910** | **900,741** | **1,000,509** | **1,111,313** |
| **Total Equity and Liabilities** | **3,521,127** | **4,451,150** | **5,098,534** | **5,843,120** | **6,706,739** | **7,703,390** | **8,848,597** | **10,159,568** | **11,655,370** | **13,357,122** |

**Figure 2.** Statement of financial position with linear business model. Source: Authors' calculations, Reporting Portal of Georgia [29], National Statistics Office of Georgia [28].

Assets:

Property, Plant, and Equipment: This category encompasses the company's tangible assets, comprising land, plant, and machinery essential for vineyard cultivation and wine production. These assets undergo depreciation based on the straight-line method, spread over a useful life span of 50 years.

Biological Assets: These represent cultivated vines, whose market value is employed for determining the valuation of grapes.

Inventories: Inventories encompass assets intended for sale, capable of generating income for the company. The annual inventory balance is strategically managed to avoid any deficits.

Trade Receivables: These are recalculated under the assumption that not all revenues recognized in the current year will materialize in the form of cash inflows. Consequently, 20% of the outstanding balances are expected to remain unpaid within the current year.

Tax Assets: Given the company's status as a Value Added Tax (VAT) payer, the VAT balance is incorporated within the tax assets section.

Equity:

Share Capital: Share capital signifies the initial investment contributed to the company at its inception. This amount is determined to ensure the coverage of capital expenditures adequately.

Profit/Loss for the Current Period: This denotes the net profit recorded at the end of the period, as presented in the profit and loss account. It encompasses all revenues and expenses.

Retained Earnings/Losses: Retained earnings encompass the cumulative net profits from previous periods, potentially available for future dividend payouts.

Liabilities:

Trade Payables: Given the company's practice of not settling all expenses accrued within the current year, payment timelines often depend on the agreed terms with suppliers. Similar to trade receivables, 20% of annual expenses are retained as unpaid.

Taxes: This category comprises sales tax expenses recognized within the current year.

Cash Flow Statement, shown in Figure 3:

| Cash Flow Statement | Year 1 | Year 2 | Year 3 | Year 4 | Year 5 | Year 6 | Year 7 | Year 8 | Year 9 | Year 10 |
|---|---|---|---|---|---|---|---|---|---|---|
| *All amounts are in EUR* | | | | | | | | | | |
| **Operational Activities** | | | | | | | | | | |
| **Cash Inflow from Operational Activities** | | | | | | | | | | |
| Sold Wine | - | 1,851,714 | 2,561,328 | 2,855,922 | 3,172,929 | 3,525,124 | 3,916,413 | 4,351,135 | 4,834,111 | 5,370,697 |
| **Total Cash Inflow from Operational Activities** | - | 1,851,714 | 2,561,328 | 2,855,922 | 3,172,929 | 3,525,124 | 3,916,413 | 4,351,135 | 4,834,111 | 5,370,697 |
| | | | | | | | | | | |
| **Cash Outflow from Operational Activities** | | | | | | | | | | |
| Purchase of Raw Materials | - | (772,535) | (888,415) | (1,021,678) | (1,174,929) | (1,351,169) | (1,553,844) | (1,786,921) | (2,054,959) | (2,363,203) |
| Salries | - | (530,958) | (557,506) | (585,381) | (614,650) | (645,382) | (677,652) | (711,534) | (747,111) | (784,466) |
| Taxes | - | (27,239) | (26,827) | (26,414) | (26,002) | (25,590) | (25,178) | (24,766) | (24,354) | (23,941) |
| *VAT* | - | - | (238,174) | (272,941) | (302,275) | (334,690) | (370,505) | (410,070) | (453,772) | (502,037) |
| Operational Expenditures | - | (195,777) | (267,110) | (295,168) | (325,624) | (359,306) | (396,559) | (437,762) | (483,338) | (533,751) |
| **Total Cash Outflow from Operational Activities** | - | (1,526,508) | (1,978,031) | (2,201,582) | (2,443,481) | (2,716,138) | (3,023,738) | (3,371,053) | (3,763,535) | (4,207,398) |
| **Net Cash from Operational Activities** | - | 325,206 | 583,297 | 654,339 | 729,448 | 808,986 | 892,675 | 980,081 | 1,070,578 | 1,163,299 |
| | | | | | | | | | | |
| **Financing Activities** | | | | | | | | | | |
| **Cash Inflow from Financing Activities** | | | | | | | | | | |
| Shareholders Equity | 3,521,127 | | | | | | | | | |
| **Total Cash Inflow from Financing Activities** | 3,521,127 | - | - | - | - | - | - | - | - | - |
| **Net Cash from Financing Activities** | 3,521,127 | - | - | - | - | - | - | - | - | - |
| | | | | | | | | | | |
| **Cash Outflow from Investment Activities** | | | | | | | | | | |
| Capital Expenditures | (2,765,095) | | | | | | | | | |
| **Tital Outflow from Investment Activities** | (2,765,095) | - | - | - | - | - | - | - | - | - |
| **Total Cash Inflow from Investment Activities** | (2,765,095) | - | - | - | - | - | - | - | - | - |
| | | | | | | | | | | |
| **Starting Balance** | - | 756,032 | 1,081,237 | 1,664,534 | 2,318,873 | 3,048,321 | 3,857,307 | 4,749,982 | 5,730,063 | 6,800,641 |
| **Cash during the period** | 756,032 | 325,206 | 583,297 | 654,339 | 729,448 | 808,986 | 892,675 | 980,081 | 1,070,578 | 1,163,299 |
| **Cash at the end of the period** | 756,032 | 1,081,237 | 1,664,534 | 2,318,873 | 3,048,321 | 3,857,307 | 4,749,982 | 5,730,063 | 6,800,641 | 7,963,940 |

**Figure 3.** Cash flow statement with linear business model. Source: Authors' calculations, Reporting Portal of Georgia [29], National Statistics Office of Georgia [28].

Operating cash flows are projected using the direct method, involving the recalibration of cash inflows and outflows based on profit and loss data, which incorporates taxes. It accounts for the fact that 20% of both accrued income and expenses remain within trade receivables and payables for the current year, with only 80% realized or settled during the same period.

Financial activities encompass the initial investment amount utilized by the company at the outset to cover capital expenditures.

Investment activities entail the capital expenditures incurred by the company in the initial year. These expenditures exclusively encompass the acquisition of land, construction of a production facility, and the procurement of machinery and equipment essential for vineyard maintenance and wine production.

Liabilities:

In light of the corporate practice of deferring certain expenses incurred within a fiscal year to subsequent periods, the timing of expenditure disbursement is often contingent upon the stipulated terms of payment negotiated with suppliers. Consequently, a portion equivalent to 20% of annual expenditures remains unpaid, resembling trade payables.

Taxation:

The statement of cash flows is prepared utilizing the direct method to prognosticate operational cash flows. Cash inflows and outflows are subsequently recalibrated in accordance with the data from the profit and loss statement, which encompasses recognized sales tax expenses for the current year. It is imperative to acknowledge that 20% of the accrued revenues and expenses persist as assets and liabilities within trade receivables and payables, while the remaining 80% is duly reflected in the contemporaneous financial reporting.

Financing Activities:

The financial activities encompass the initial investment outlay with which the company commenced its operations and facilitated coverage of capital expenditures.

Investment Activities:

The investment activities primarily encapsulate the capital expenditures incurred by the company in its inaugural year. This pertains exclusively to the acquisition of land, construction of a production facility, and the procurement of machinery and equipment requisite for the maintenance of the vineyard and the production of wine.

### 3.2.2. Circular Business Model

In accordance with the tenets of the circular business model, congruent with the conventional linear business model, the company engages in wine production while concurrently deriving value from the utilization of grape waste, thus generating revenue streams from both wine bottle sales and the commercialization of grape seed oil. This operational paradigm persists under the framework of a 10-million-dollar investment, mirroring the structure of the antecedent business model.

Profit and Loss Statement, shown in Figure 4:

| Profit and Loss Statement All amounts are in EUR | Year 1 | Year 2 | Year 3 | Year 4 | Year 5 | Year 6 | Year 7 | Year 8 | Year 9 | Year 10 |
|---|---|---|---|---|---|---|---|---|---|---|
| **Revenue** | - | 2,128,993 | 2,410,026 | 2,680,389 | 2,981,111 | 3,315,603 | 3,687,662 | 4,101,510 | 4,561,847 | 5,073,900 |
| *Wine* | | 1,898,007 | 2,150,859 | 2,389,605 | 2,654,851 | 2,949,539 | 3,276,938 | 3,640,678 | 4,044,794 | 4,493,766 |
| *Grape Oil* | | 230,986 | 259,166 | 290,784 | 326,260 | 366,064 | 410,724 | 460,832 | 517,054 | 580,134 |
| COGs Wine | - | (1,005,149) | (1,101,017) | (1,207,243) | (1,325,024) | (1,455,698) | (1,600,766) | (1,761,906) | (1,940,997) | (2,140,144) |
| COGs Oil | - | (4,666) | (5,235) | (5,874) | (6,590) | (7,394) | (8,297) | (9,309) | (10,444) | (11,719) |
| **Gross Profit** | - | 1,119,178 | 1,303,773 | 1,467,272 | 1,649,497 | 1,852,511 | 2,078,599 | 2,330,296 | 2,610,406 | 2,922,037 |
| GP, % | 0% | 53% | 54% | 55% | 55% | 56% | 56% | 57% | 57% | 58% |
| | | | | | | | | | | |
| Salaries and Other Employee Benefits | - | (159,287) | (167,252) | (175,614) | (184,395) | (193,615) | (203,295) | (213,460) | (224,133) | (235,340) |
| Sales and Marketing Expenses | - | (42,580) | (48,201) | (53,608) | (59,622) | (66,312) | (73,753) | (82,030) | (91,237) | (101,478) |
| Transportation Expenditures | - | (21,290) | (24,100) | (26,804) | (29,811) | (33,156) | (36,877) | (41,015) | (45,618) | (50,739) |
| Consulting Expenses | - | (21,290) | (24,100) | (26,804) | (29,811) | (33,156) | (36,877) | (41,015) | (45,618) | (50,739) |
| Insurance | - | (18,217) | (20,622) | (22,935) | (25,508) | (28,370) | (31,554) | (35,095) | (39,034) | (43,415) |
| Fuel | - | (14,861) | (16,823) | (18,710) | (20,809) | (23,144) | (25,741) | (28,630) | (31,844) | (35,418) |
| Utilities | - | (89,853) | (98,401) | (107,804) | (118,147) | (129,524) | (142,039) | (155,806) | (170,949) | (187,606) |
| Depreciation and Amortization | - | (42,978) | (42,978) | (42,978) | (42,978) | (42,978) | (42,978) | (42,978) | (42,978) | (42,978) |
| Taxes | - | (28,101) | (27,672) | (27,242) | (26,812) | (26,382) | (25,953) | (25,523) | (25,093) | (24,663) |
| Other OPEX | - | (6,435) | (6,894) | (7,367) | (7,877) | (8,429) | (9,025) | (9,670) | (10,369) | (11,127) |
| **Operating Revenue** | - | 674,285 | 826,731 | 957,407 | 1,103,726 | 1,267,444 | 1,450,507 | 1,655,073 | 1,883,532 | 2,138,534 |
| | | | | | | | | | | |
| Profit Tax | - | (6,743) | (8,267) | (9,574) | (11,037) | (12,674) | (14,505) | (16,551) | (18,835) | (21,385) |
| **Net Income** | - | 667,542 | 818,464 | 947,833 | 1,092,689 | 1,254,770 | 1,436,002 | 1,638,522 | 1,864,697 | 2,117,148 |
| NI, % | 0% | 31% | 34% | 35% | 37% | 38% | 39% | 40% | 41% | 42% |

**Figure 4.** Profit and loss statement with circular business model. Source: Authors' calculations, Reporting Portal of Georgia [29], National Statistics Office of Georgia [28].

Revenue from Sales: Commencing its investment activities in the inaugural year, and as previously expounded, sales figures are not yet determinate. As such, sales initiation occurs in the second year of operation, in accordance with anticipated yields. From a total vineyard area of 200 hectares, we anticipate a production output of 1,000,000 standard wine bottles (with a capacity of 0.75 L) in the second year. Based on researched market data, these wine bottles are valued at $2.5 USD [30] at the prevailing exchange rate of 6.30 Georgian Lari, thereby generating an income of 5.4 million GEL for the company in the second year. In addition to wine sales, the company engages in the extraction of rapeseed oil through the processing of grape waste. According to research findings, 10 kg of residual pomace, remaining from every 500 kg of grapes, suffices for the production of 1 L of grape seed oil. Concomitantly, the company, adhering to the prescribed business model, procures grape residues from local entities at a rate of 20 GEL per kilogram to facilitate oil production. Based on these stipulated assumptions, the second year yields 20,000 bottles of grape seed oil (0.2-L bottles) for sale, with an initial price of 40 GEL per bottle. Thereafter, sales volumes for both wine and oil increase by 10% annually, in line with the growth in wine sales, and pricing escalates by 2%.

Cost of Goods Sold: The cost of producing a wine bottle encompasses the expense of 1 kg of Rkatshiteli grapes, packaging materials, and the salary of a factory worker directly involved in the production process. The grape cost is commensurate with market prices and is assumed to undergo an annual increment of 2% congruent with the sales price escalation. Expenses related to bottle production, labels, thermocaches, and stoppers are also pegged to market prices [27] with a projected annual upswing of 2% in synchronization with the sales price. Regarding labor costs, in light of the employment of 70 personnel within the enterprise, 70% of the total salary expenditure is apportioned to the cost category, while the remaining 30% is allocated to operating expenses. The cost structure for grape oil comprises the market rates of packaging materials and the cost of waste transportation. This cost is calculated as 2% of the grape yield from 1 kg, approximating 0.02 GEL, and is projected to increment by 2% each successive year in correspondence with sales price fluctuations.

Operating Expenses: Salary expenditures are derived from the average monthly salary of 2018–2022 data, amounting to 1176 GEL [26]. These costs are further compounded by an annual escalation of 5%. Marketing, transportation, consulting, insurance, and fuel expenses are evaluated as a percentage of sales revenue, encompassing 1–2%. Annual utility costs are computed in compliance with local tariffs and the sanctioned production consumption, factoring in water and electricity usage. Based on researched information, irrigating 1 hectare necessitates an average of 2 million liters of water, whereas the production of 1 L of wine consumes 4.74 L of water. In terms of electric energy, bottling 1 million bottles entails a consumption of 22,540 kWh, adjusted and calculated in tandem with annual sales projections.

Depreciation: Depreciation is computed using the straight-line method, with long-term assets amortized over an estimated useful life of 50 years.

Taxes: Taxation includes property tax at a rate of 1% of the value of fixed assets.

Unforeseen Costs: Given the speculative nature of the financial model presented, the company may be confronted with unforeseen expenses, which are conservatively projected at 3% of sales.

Income and Expenses Net of Taxes: All figures presented in the financial model are depicted net of taxes.

Financial Statement, shown in Figure 5:

| Statement of Financial Position | Year 1 | Year 2 | Year 3 | Year 4 | Year 5 | Year 6 | Year 7 | Year 8 | Year 9 | Year 10 |
|---|---|---|---|---|---|---|---|---|---|---|
| *All amounts are in EUR* | | | | | | | | | | |
| **Assets** | | | | | | | | | | |
| Fixed Assets | 2,430,588 | 2,430,588 | 2,430,588 | 2,430,588 | 2,430,588 | 2,430,588 | 2,430,588 | 2,430,588 | 2,430,588 | 2,430,588 |
| Biological assets | 422,535 | 422,535 | 422,535 | 422,535 | 422,535 | 422,535 | 422,535 | 422,535 | 422,535 | 422,535 |
| *Accumulated Depreciation* | - | (42,978) | (85,956) | (128,934) | (171,912) | (214,890) | (257,868) | (300,846) | (343,824) | (386,802) |
| Inventory | - | - | 22,269 | 72,131 | 156,683 | 284,449 | 465,634 | 712,432 | 1,039,373 | 1,463,740 |
| Accounts Receivable | - | 519,267 | 587,811 | 653,753 | 727,100 | 808,684 | 899,430 | 1,000,368 | 1,112,646 | 1,237,537 |
| Prepaid TAX | - | 187,172 | 209,665 | 234,303 | 261,866 | 292,701 | 327,199 | 365,796 | 408,980 | 457,301 |
| Cash and Cash Equivalents | 668,004 | 1,198,491 | 2,015,844 | 2,932,938 | 3,957,339 | 5,097,405 | 6,361,696 | 7,758,874 | 9,297,579 | 10,986,263 |
| **Total Assets** | **3,521,127** | **4,715,075** | **5,602,757** | **6,617,315** | **7,784,200** | **9,121,472** | **10,649,214** | **12,389,748** | **14,367,878** | **16,611,162** |
| | - | - | - | - | - | - | - | - | - | - |
| **Shareholder Equity** | - | - | - | - | - | - | - | - | - | - |
| Share capital | 3,521,127 | 3,521,127 | 3,521,127 | 3,521,127 | 3,521,127 | 3,521,127 | 3,521,127 | 3,521,127 | 3,521,127 | 3,521,127 |
| Current Period Profit (loss) | - | 667,542 | 818,464 | 947,833 | 1,092,689 | 1,254,770 | 1,436,002 | 1,638,522 | 1,864,697 | 2,117,148 |
| Accumulated Profit (loss) | - | - | 667,542 | 1,486,006 | 2,433,838 | 3,526,527 | 4,781,297 | 6,217,299 | 7,855,821 | 9,720,518 |
| **Total Equity** | **3,521,127** | **4,188,669** | **5,007,132** | **5,954,965** | **7,047,654** | **8,302,423** | **9,738,426** | **11,376,948** | **13,241,645** | **15,358,793** |
| | | | | | | | | | | |
| **Liabilities** | - | - | - | - | - | - | - | - | - | - |
| Accounts Payable | - | 52,323 | 58,327 | 64,398 | 71,119 | 78,559 | 86,797 | 95,918 | 106,017 | 117,201 |
| TAX Payables | - | 474,083 | 537,297 | 597,952 | 665,428 | 740,490 | 823,992 | 916,882 | 1,020,216 | 1,135,168 |
| **Total Liabilities** | **-** | **526,406** | **595,624** | **662,350** | **736,546** | **819,049** | **910,788** | **1,012,800** | **1,126,233** | **1,252,369** |
| **Total Equity and Liabilities** | **3,521,127** | **4,715,075** | **5,602,757** | **6,617,315** | **7,784,200** | **9,121,472** | **10,649,214** | **12,389,748** | **14,367,878** | **16,611,162** |

**Figure 5.** Financial statement with circular business model. Source: Authors' calculations, Reporting Portal of Georgia [29], National Statistics Office of Georgia [28].

Assets:

The category of fixed assets encompasses the company's real properties, inclusive of land, manufacturing facilities, and machinery requisite for vineyard cultivation, wine production, and oil manufacturing. Depreciation of these assets follows a linear amortization method, allocated over a useful life of 50 years. Furthermore, a biological asset, representing cultivated vines, contributes to the valuation of grapes in the inventory.

Inventories:

Inventories consist of assets earmarked for sale, generating revenue for the company. In this context, inventories encompass wine and oil stocks. The annual inventory balance is meticulously planned to prevent any shortfalls.

Trade Claims:

Trade claims are recalculated with the assumption that revenues recognized in the present fiscal year will not entirely manifest as cash flows to the company. Consequently, a proportion equivalent to 20% of the balances remains outstanding during the current year.

Tax Assets:

The company, being a value-added tax (VAT) payer, accounts for VAT balances within the tax assets section.

Equity:

Share capital represents the initial investment injected into the company upon its establishment. This sum is computed in alignment with the reimbursement of capital expenditures incurred in the production of wine and oil.

Profit/Loss for the Current Period:

This figure reflects the net profit at the conclusion of the period, calculated subsequent to a comprehensive consideration of all revenues and expenses, inclusive of the gross profit derived from oil sales.

Retained Earnings/Losses:

Retained earnings encompass the cumulative net profits accrued from preceding periods, serving as a reservoir from which potential dividends may be disbursed in the future.

Liabilities:

This financial model accommodates the operational reality that the company does not cover all expenses incurred within the current fiscal year during the same period. Accordingly, akin to trade claims, a 20% portion of the annual expenses remains unpaid.

Taxes:

The tax category encompasses the expense accrued from sales tax recognized within the present fiscal year. The company does not bear any other obligations.

Cash flow statement, shown in Figure 6:

| Cash Flow Statement | Year 1 | Year 2 | Year 3 | Year 4 | Year 5 | Year 6 | Year 7 | Year 8 | Year 9 | Year 10 |
|---|---|---|---|---|---|---|---|---|---|---|
| All amounts are in EUR | | | | | | | | | | |
| Operational Activities | | | | | | | | | | |
| Cash Inflow from Operational Activities | | | | | | | | | | |
| Sold Wine | - | 1,851,714 | 2,561,328 | 2,855,922 | 3,172,929 | 3,525,124 | 3,916,413 | 4,351,135 | 4,834,111 | 5,370,697 |
| Sold Grape Oil | | 225,352 | 309,183 | 346,903 | 389,226 | 436,711 | 489,990 | 549,769 | 616,841 | 692,095 |
| Total Cash Inflow from Operational Activities | - | 2,077,066 | 2,870,511 | 3,202,825 | 3,562,155 | 3,961,835 | 4,406,403 | 4,900,903 | 5,450,951 | 6,062,792 |
| Cash Outflow from Operational Activities | - | | | | | | | | | |
| Purchase of Raw Materials | - | (778,225) | (895,438) | (1,029,557) | (1,183,770) | (1,361,088) | (1,564,974) | (1,799,408) | (2,068,970) | (2,378,923) |
| Salries | - | (530,958) | (557,506) | (585,381) | (614,650) | (645,382) | (677,652) | (711,534) | (747,111) | (784,466) |
| Taxes | - | (28,101) | (27,672) | (27,242) | (26,812) | (26,382) | (25,953) | (25,523) | (25,093) | (24,663) |
| VAT | - | - | (286,911) | (327,633) | (363,649) | (403,561) | (447,789) | (496,793) | (551,087) | (611,236) |
| Operational Expenditures | - | (209,294) | (285,632) | (315,919) | (348,872) | (385,354) | (425,745) | (470,467) | (519,985) | (574,819) |
| Total Cash Outflow from Operational Activities | - | (1,546,578) | (2,053,158) | (2,285,732) | (2,537,754) | (2,821,769) | (3,142,112) | (3,503,725) | (3,912,246) | (4,374,108) |
| Net Cash from Operational Activites | - | 530,488 | 817,353 | 917,093 | 1,024,401 | 1,140,066 | 1,264,291 | 1,397,178 | 1,538,705 | 1,688,684 |
| Financing Activities | | | | | | | | | | |
| Cash Inflow from Financing Activities | - | | | | | | | | | |
| Shareholders Equity | 3,521,127 | | | | | | | | | |
| Total Cash Inflow from Financing Activities | 3,521,127 | - | - | - | - | - | - | - | - | - |
| Net Cash from Financing Activities | 3,521,127 | - | - | - | - | - | - | - | - | - |
| Cash Outflow from Investment Activities | | | | | | | | | | |
| Capital Expenditures | (2,853,123) | | | | | | | | | |
| Total Cashflow from Investment Activities | (2,853,123) | - | - | - | - | - | - | - | - | - |
| Net Cash from Investment Activities | (2,853,123) | - | - | - | - | - | - | - | - | - |
| Starting Balance | - | 668,004 | 1,198,491 | 2,015,844 | 2,932,938 | 3,957,339 | 5,097,405 | 6,361,696 | 7,758,874 | 9,297,579 |
| Cash during the period | 668,004 | 530,488 | 817,353 | 917,093 | 1,024,401 | 1,140,066 | 1,264,291 | 1,397,178 | 1,538,705 | 1,688,684 |
| Cash at the end of the period | 668,004 | 1,198,491 | 2,015,844 | 2,932,938 | 3,957,339 | 5,097,405 | 6,361,696 | 7,758,874 | 9,297,579 | 10,986,263 |

**Figure 6.** Source of cash flow with circular business model. Source: Authors' calculations, Reporting Portal of Georgia [29], National Statistics Office of Georgia [28].

The direct method is used for the forecast of operating cash flows, cash inflows and outflows are calculated according to profit and loss data (including taxes), taking into account the fact that 20% of accrued income and expenses remained in trade receivables and payables in the current year and only 80% was credited/transferred in the current period.

The financial activity includes the investment amount with which the company started operating and covered the capital expenditures.

Investment activities include the company's capital expenditures incurred in the first year, including only the purchase of land, the construction of a production plant, and the cost of machinery and equipment required for vineyard maintenance and wine production.

Comparative Analysis of Financial Indicators

In the assessment of the investment project's profitability, various financial metrics, including Net Present Value (NPV), Internal Rate of Return (IRR), Payback Period (PBP), and Profitability Index (PI), are computed on the foundation of empirical findings. Isolated evaluation of these individual indices in isolation does not provide a comprehensive basis for rendering judgment on the overall profitability of a given project. Therefore, it becomes imperative to engage in a systematic analysis and comparative examination of the data derived from these indicators in tandem, see Figure 7.

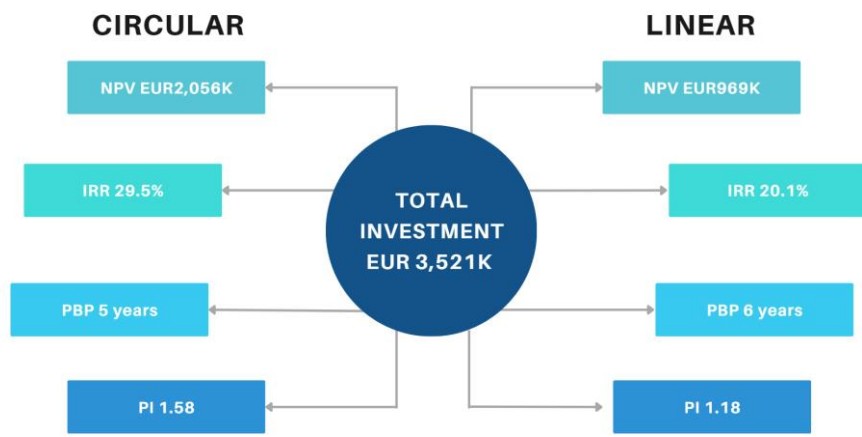

**Figure 7.** Coefficients of the Investment Project: Circular vs Linear. Source: Authors' calculations.

Financial indicators play a pivotal role in assessing the viability and profitability of investment projects. Four key indices, namely Net Present Value (NPV), Internal Rate of Return (IRR), Payback Period (PBP), and Profitability Index (PI), are employed to provide a comprehensive evaluation. The interpretation of these indicators is contingent upon

the specific industry and the project's characteristics, and as such, there is no universally prescribed threshold; however, generally speaking, favorable assessments manifest as positive values.

A cursory examination of the table reveals that the investment project developed under the aegis of a circular business model yields a positive NPV, a desired outcome in the assessment of any investment endeavor. In parallel, the linear model also produces a positive NPV; nonetheless, when compared to the circular model, the magnitude of the NPV is considerably lower. This discrepancy signifies that while the linear model is projected to be financially remunerative, it is characterized by a heightened degree of financial risk relative to the circular model.

Turning attention to the IRR indicator, a measure denoting the rate at which the present values of cash inflows and outflows equate, the circular business model presents a favorable IRR of 29.5%. This value suggests a financially promising investment prospect. In the case of the linear business model, the IRR is calculated at 20.1%, denoting that the current value of cash flow inflows exceeds that of outflows, thereby affirming the profitability of the investment project.

The Payback Period (PBP) is a straightforward measure delineating the time required to recoup investment outlays with generated cash flows. For the linear model, the PBP is estimated at the 6th year, signifying a six-year duration for the full recovery of investment costs. In contrast, the circular model exhibits a PBP of 5 years, indicating a one-year advantage in terms of recouping the initial investment. Although there exists a marginal difference in the PBP between the two models, the circular business model stands as the swifter investment in terms of cost recovery.

The Profitability Index (PI) represents the ratio of the present value of future cash flows generated by the investment to the initial investment. It serves as an informative metric for assessing the profitability, risk profile, and financial efficacy of a project. An index value less than 1 suggests potential future losses, while a value greater than 1 signifies profitability. In this case, the circular model yields a PI of 1.58, indicating that the present value of future cash flows is 1.58 times greater than the initial investment. Conversely, the linear model registers a PI of 1.18, implying that for each unit of currency invested, the present value of cash flows is equivalent to 1.18 currency units. The circular model thus demonstrates a superior profitability index, rendering it a more attractive proposition for potential investors.

In summation, these four key indices facilitate an objective appraisal of investment projects. The findings herein suggest that, among two projects subject to identical investment conditions, the circular business model exhibits superior profitability compared to the traditional linear model.

Profitability Ratios

The Gross Profit Margin, a critical profitability metric, elucidates the proportion of a company's sales revenue that constitutes the total profit generated. A higher value in this ratio signifies a substantial difference between the selling price of the company's product and its cost price, thereby reflecting an efficient cost management system, see Figure 8.

Net Profit Margin: An Indicator of Financial Performance

The Net Profit Margin, an essential profitability ratio, is determined through the ratio of net profit to sales revenue, providing insight into the proportion of net profit in relation to overall sales. A higher net profit margin denotes a heightened level of financial stability within the company. These metrics furnish the basis for assessing the company's adeptness in cost management and its capacity to generate returns for every unit of currency invested. Notably, net profit, distinct from gross profit, is calculated subsequent to the accounting for all incurred costs, resulting in a margin figure inherently lower than the gross profit margin.

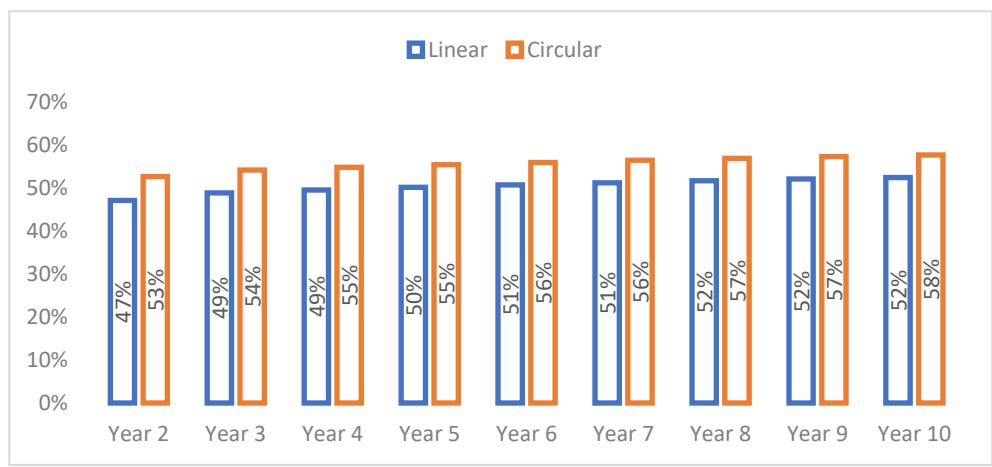

**Figure 8.** Gross profit margin comparison (Circular vs. Linear). Source: Authors' calculations, Reporting Portal of Georgia [29], National Statistics Office of Georgia [28].

Comparing the net profit margin indicators of the two models reveals a semblance to the differential observed in the gross profit ratios. It is noteworthy that the net profit derived from the circular business model encompasses revenue from grape seed oil sales and ancillary operating expenses. Nevertheless, the outcome remains superior, signifying a more judicious financial performance, notwithstanding the inclusion of additional operational costs in the calculation, see Figure 9.

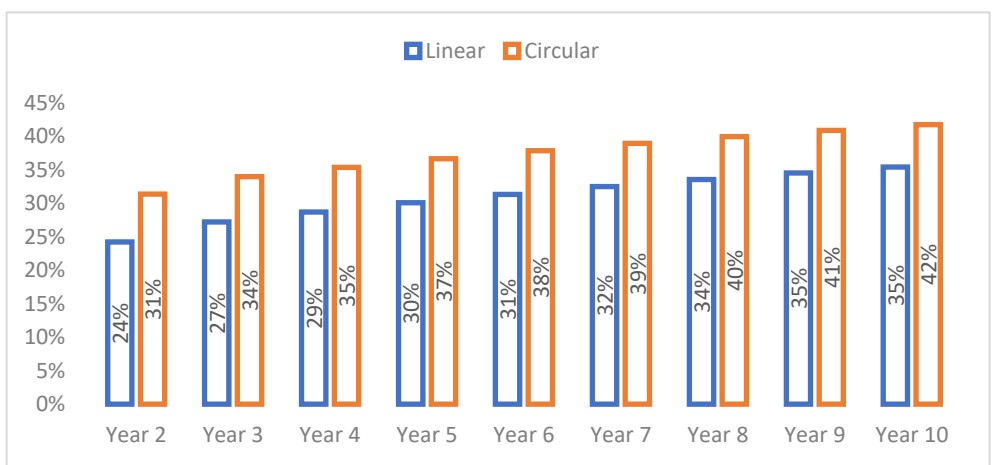

**Figure 9.** Net profit margin comparison (Circular vs. Linear). Source: Authors' calculations, Reporting Portal of Georgia [29], National Statistics Office of Georgia [28].

Return on equity (ROE) measures how efficiently a company uses capital to generate profits, calculated as the ratio of net income to capital, see Figure 10. A higher rate of return on capital means that the enterprise earns more profit for each EURO invested and uses capital efficiently.

Over 10 years, the ratio of both models is increasing, which is the result of annual sales growth, but the given financial indicators are quite different from each other, which is naturally not surprising after analyzing the net profit margin data.

A high circular model ROE is an indication that profitability and shareholder wealth generation is increasing. A company operating with a circular business model is able to generate more profit over 10 years, considering that the share capital is the same in both models.

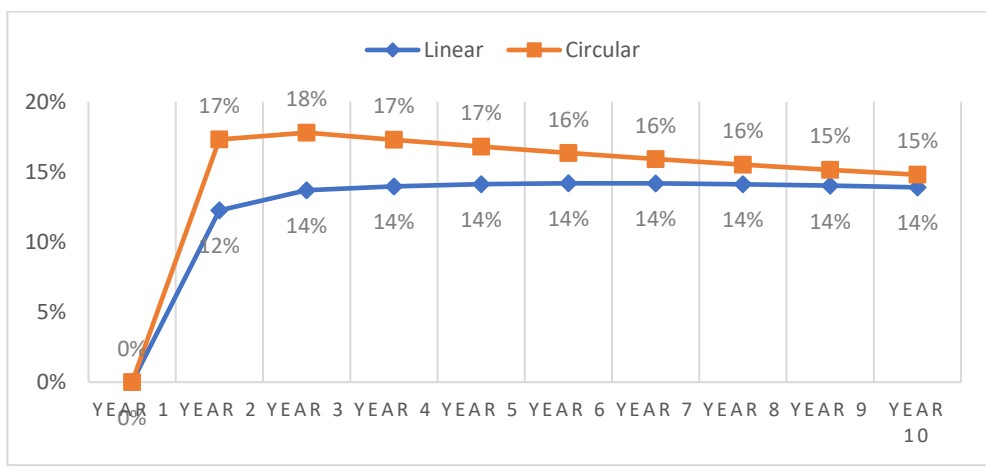

**Figure 10.** Return on Equity (Circular vs. Linear). Source: Authors' calculations, Reporting Portal of Georgia [29], National Statistics Office of Georgia [28].

Return on Assets (ROA): A Measure of Profitability Relative to Assets

The Return on Assets (ROA) is a key financial metric derived from the ratio of net profit to total assets. This metric illuminates the company's capacity to generate profits with its existing assets, essentially depicting the efficiency with which the company employs its asset base to yield profitability.

In the context of the linear business model, there is an observable upward trajectory in the ROA. Conversely, in the circular business model, the ROA experiences a marginal decrement, see Figure 11.

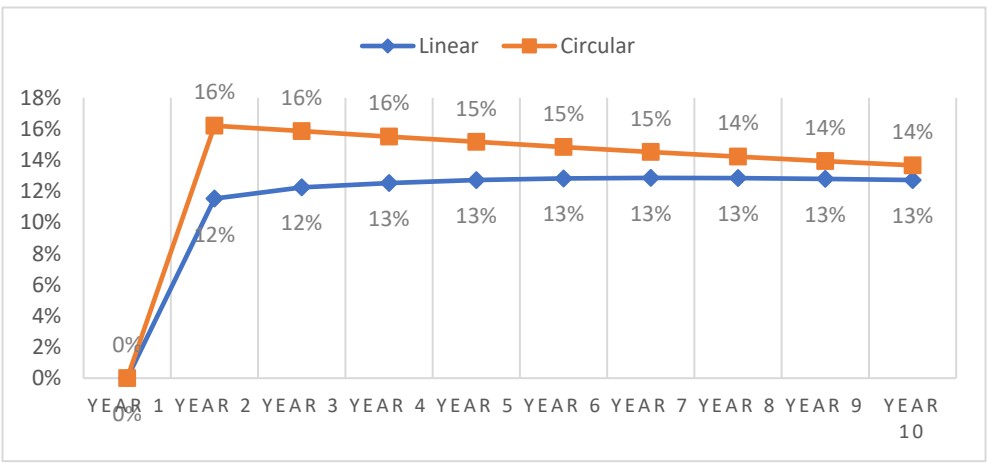

**Figure 11.** Return on Assets (Circular vs. Linear). Source: Authors' calculations, Reporting Portal of Georgia [29], National Statistics Office of Georgia [28].

Although the discrepancies in the ROA ratios are relatively modest, even this marginal variance imparts a consequential inference: the circular business model, from a financial standpoint, demonstrates superior efficacy in leveraging its extant assets to yield profitability when compared to the linear business model.

Net cash from operating activities represents the disparity between cash inflows and outflows stemming from a company's fundamental business operations. These operating activities encompass pivotal functions such as sales, procurement, and operational expenditures. Operating cash flow serves as a vital component within financial data, shedding light on the capacity of a company to generate cash from its routine, day-to-day operations.

In the context of our specific illustrations, it is discernible that both models exhibit positive net cash from operating activities, except for the initial year when sales were not

recorded. This observation indicates that, generally, both models are accruing more cash through their sales endeavors than they are disbursing. This dynamic corroborates the premise that the company's operations are profitable.

Further examination of the net operating cash values from both models, computed positively across nearly every year, offers insights into which of the models garners a superior yield from their principal business activities. The graphical representation also underscores the fact that the project devised under the circular business model consistently retains a greater surplus of operating cash when contrasted with the linear model. This outcome is logical, as the primary distinction between the linear and circular models pertains to the sale of grape seed oil, which augments supplemental cash flow, shown in Figure 12.

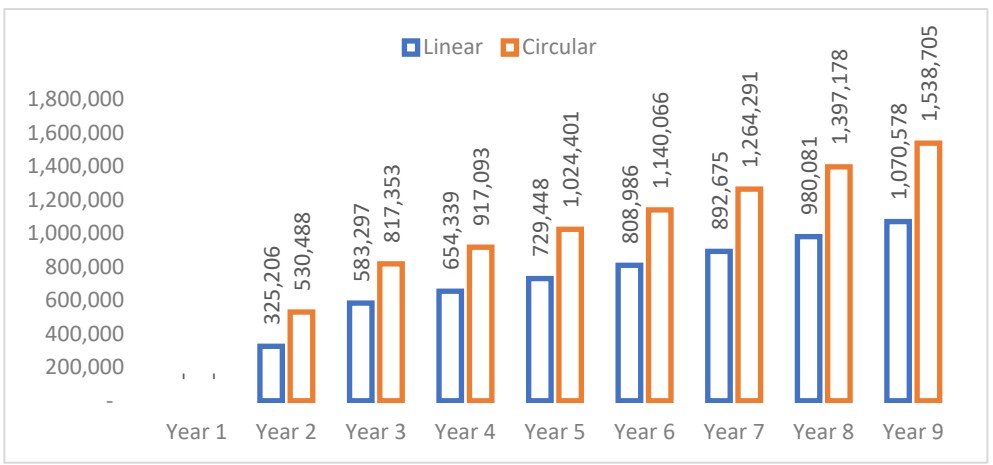

**Figure 12.** Net cash from operating activities (Circular vs. Linear). Source: Authors' calculations, Reporting Portal of Georgia [29], National Statistics Office of Georgia [28].

### 4. Discussion

The findings of this research transcend regional boundaries and can be applied universally. The issue at hand, along with the proposed solutions, is not confined by geographical constraints. As a result, these research outcomes, with due consideration for their assumptions and approaches, present a valuable resource for both international markets and local enterprises contemplating the transition from the linear, traditional business model to a contemporary circular business model.

Moreover, the potential for generalization extends to a myriad of industries, encompassing agriculture, healthcare, industry, energy, and more. Industries grappling with surplus waste generation, akin to the wine sector, have the potential to adapt and transform their production processes into a circular business model, thereby engendering added value and fostering a more sustainable and prosperous future.6. Patents

Future research endeavors can explore an array of methodologies, building upon the literature review. Potential directions include the introduction of a circular business model for wastewater recycling, thereby reducing long-term operating costs for wine enterprises. Additionally, research could extend into the development of energy and cosmetics production as supplementary income sources for wine companies. The conversion of grape waste into organic fertilizers represents another environmentally responsible avenue worth exploring.

### 5. Conclusions

In conclusion, this comprehensive analysis, informed by a review of existing literature, an in-depth exploration of the wine industry, dedicated research efforts, and a careful evaluation of the findings, allows us to draw evidence-based conclusions regarding the topic at hand.

1. Environmental Sustainability: The imperatives of the modern world underscore the urgency of embracing environmental sustainability as a primary objective across all industries. The linear business model, emblematic of traditional practices, contributes to environmental pollution through waste generation. However, the current landscape offers boundless opportunities and technologies for enterprises to refine their production methods, align them with ecological considerations, and achieve a confluence of operational and financial benefits.

2. Waste Transformation: Agricultural surplus, especially within the wine industry, holds the potential to be transformed into valuable products that enhance both production processes and financial performance. Our research underscores that a circular business model, operating with equivalent capital investment, is poised to deliver superior financial outcomes.

3. Innovative Solutions: Machines designed for grape residue processing, characterized by moderate capital costs, offer a rapid return on investment. This trend is further bolstered by the growing demand for organic products in sectors such as cosmetics, healthcare, and food.

4. Pioneering Initiatives: The nascent state of waste processing in the Georgian market positions the proposed project as innovative and conducive to local economic growth and environmental conservation.

Recommendations for Action:

1. Environmental Awareness: Fostering greater social responsibility and heightened awareness of environmental issues is paramount. Recognizing the potential to enact positive change through daily activities is vital in the contemporary context.

2. Value Addition: Wine companies should strive to create value-added products that curtail waste disposal, concurrently addressing environmental and financial concerns. This entails the adoption of cutting-edge technologies, a cornerstone of the circular business model, particularly through the recycling of grape waste and wastewater.

3. Financial Benefits: The recurring lost revenues, consigned to landfills under the circular business model, accentuate the rationale for its adoption. The research findings unequivocally demonstrate that, under equivalent capital investment, the circular model consistently yields superior financial returns and indicators, thereby enhancing financial stability and sustainability.

4. Collaboration and Engagement: The envisioned grape waste collection from local enterprises, integral to the circular investment plan, presents an opportunity to foster engagement and interest, not only from a financial perspective but also in terms of technological and business environment development.

**Author Contributions:** Conceptualization, V.C., D.A. and G.D.; methodology, V.C.; software, G.D.; validation, D.A. and R.S.; formal analysis, V.C.; investigation, G.D.; resources, R.S.; data curation, D.A.; writing—original draft preparation, G.D.; writing—review and editing, V.C. and D.A.; visualization, D.A.; supervision, D.A. and R.S.; project administration, V.C.; funding acquisition, D.A. All authors have read and agreed to the published version of the manuscript.

**Funding:** Latvian Council of Science, project Quadruple Helix Concept as base of the next generation PPP model: lzp-2020/1-0062.

**Institutional Review Board Statement:** Not applicable.

**Informed Consent Statement:** Informed consent was obtained from all subjects involved in the study.

**Data Availability Statement:** Data available upon request.

**Conflicts of Interest:** The authors declare no conflict of interest.

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
