# Peer review of "Assessing the Financial Viability and Sustainability of Circular Business Models in the Wine Industry: A Comparative Analysis to Traditional Linear Business Model—Case of Georgia"

_sustainability, doi:10.3390/su16072877_

Round 1

Reviewer 1 Report

Comments and Suggestions for Authors

1. The Methods section should come after the literature review.

2. The method section is too weak and needs more elaboration on research design, measures, population, and sampling.

3. Data analysis techniques are poorly employed.

4. Tables are in pictorial form and must be provided in editable format.

5. Figures are not clear, they should be in transparent color.

6. Missing research questions, and hypothesis, how results were drawn.

7. For the review of the literature no model is specifically used.

Comments on the Quality of English Language

Needs moderate improvement

Author Response

Dear Reviewer,

Thank you for your constructive feedback. We have implemented the following changes in response to your comments:

1. **Methods Section Position**: The Methods section has been relocated to immediately follow the Literature Review, ensuring a logical flow in the manuscript.

2. **Strengthening the Method Section**
3. **Improvement in Data Analysis Techniques**: We have revised the methodology section to better elucidate our approach, which involves a mixed-method analysis combining quantitative financial analysis and qualitative assessments. This approach was chosen to provide a holistic view of the circular and linear business models in the Georgian wine industry. We understand your suggestion regarding the use of case studies. However, our approach was chosen to allow for a broader analysis of the industry as a whole, rather than focusing on individual companies. This was deemed more appropriate for our objective of providing a comprehensive overview of the industry-wide transition to circular business models.

4. **Clarity of Figures**: The figures have been revised with transparent backgrounds to improve clarity and visual appeal.

5. **Inclusion of Research Questions and Hypothesis**: Research questions and hypotheses have been clearly stated in the introduction, providing a framework for how the results were derived.

6. **Literature Review Model**: A specific model for the literature review has been identified and described, to better structure our analysis of existing research.

We appreciate the opportunity to enhance our manuscript based on your valuable insights.

Best regards,
Vakhtang

Reviewer 2 Report

Comments and Suggestions for Authors

1) The paper addresses an interesting topic regarding the evaluation of the financial viability and sustainability of circular and linear business models in the Georgian wine industry by conducting a comparative analysis between them, which resulted in the importance and predominant role of the circular business model in achieving performance financial and sustainability in the wine industry.

2) The paper is properly structured, is done with originality by the authors and corresponds to the topic chosen for the special issue of the magazine.

  3) By comparison with other studies published in this field, the element of novelty is the comparative analysis between the two business models according to an own conception and interpretation, which can be considered relevant for the scientific literature in this field.

4) Regarding the presented methods and materials section, I consider it irrelevant that what should be done in this study was exposed. It is necessary to reformulate this section by presenting the materials and methods used in carrying out this research (Partially found in the results section).

5) The results obtained and presented are relevant, clear and supported by correct arguments.

6) The discussion section presents the research carried out on the basis of the presented study and the possibility of application at the international level, as well as future research perspectives in this field for aspects that were not addressed through this study.

7) The conclusions presented are relevant to the chosen field and are consistent with the results obtained, of interest to interested readers, and the recommendations are relevant and to be considered for the evolution and development of the wine industry.

8) The references cited in the paper are recent publications from the last 5 years and relevant in the circular economy field, but I consider them insufficient. Supplementation of these references is needed.

9) The information and data presented in tables, figures, diagrams, infographics have not mentioned the source. Also, set a unique name for: figures, charts, infographic (one of these for all presentations of the same kind).

In line 43, at the beginning of the sentence there is the letter T that needs to be deleted.

The numbering of section 2. Theoretical Framework (Literature Review) should be changed to 3, respectively: 3.1, 3.2,.......

Author Response

References were added and updated. Thank you.

Reviewer 3 Report

Comments and Suggestions for Authors

The article deals with contemporary topics. A very substantial comparative analysis is presented in the paper, which can be considered a scientific contribution, and the results find their applicability in practice, not only in Georgia, but even wider than that. The abstract has slightly more words (85) than required by the journal's instructions (200). Perhaps it would be more common if the abstract were written "in a block", i.e., without separating paragraphs. The recommendation to the authors is to remove the word comprehensive from the abstract (literature review), because it is not in accordance with only 13 listed references. All references are relevant and recent, and it can be assumed that adequate literature was consulted, but certainly the word comprehensive does not correlate with the number of cited references. 

It is necessary to indicate in the integral text the references under the ordinal number in square brackets (see the instructions). In this way of referencing, all authors will be listed, not only the first on the paper, as in (Oliviera, 2016) or (Rodrigues, 2022)...etc. 

The first 6 references are incomplete. In some it says descriptively, with the words Volume, Page range, but that is missing.  In the second and eleventh references, it is written descriptively, with the words Publisher, Location, Country (that's what's missing!)

With references 7, 8 and 13 there is no time reference (when the site was accessed or when the site was updated). Likewise, with certain references, the year of publication is immediately after the author's name, which is not in accordance with the journal's instructions. Also, check where it is necessary to bold the year of publication. 

Author Response

Thank you for your positive feedback.

Reviewer 4 Report

Comments and Suggestions for Authors

Dear authors,

I read your article with interest. Indeed, I find the focus of your manuscript (waste prevention in the wine industry) very exciting and therefore think that the scope of your study would be fitting for the Sustainability Journal. However, while reading it, I noticed many points that argue against recommending publication of the article in its current form. Overall, I believe that the study needs to be completely restructured and redesigned before it is resubmitted for peer review. Starting points for the revision are summarized in my report. 

I wish you all the best for the future!

At the beginning of the article, too little reference is made to current scientific research. Only a single reference is used in the entire introductory chapter. I am also not convinced by the objective of the study. If I have understood you correctly, you are aiming for a comparison between the linear and the circular business model, whereby you are primarily interested in a comparison of the financial profitability of the two business models. To be honest, the outcome of this analysis is clear to the reader from the outset. Of course, the additional economic utilization of waste products creates additional value, which has a positive impact on overall profitability. For me, the result of the analysis is therefore predetermined from the outset. Perhaps you should better explain to the reader why this result is interesting. Would you have expected a different finding? Or is there no understanding in the wine industry of the higher profitability of the circular model? If so, how can your results help to improve the situation? 

There is a technical description at the top of page 3 which is not part of the study. This should be removed. 

Section 3 is not what I would call a theoretical framework, as no theoretical foundations are presented there. Instead, the basic conceptual rationale is explained here, which is not wrong in my opinion. I would just change the wording. Also, as in the introduction, I have the feeling that the literature has been reviewed and processed too narrowly. I also miss cross-comparisons with other industries in which the circular model has been successfully implemented. Perhaps the overview could be expanded to show how important waste recycling has become in other industries and parts of the world. Also, and I must make this very clear, the study simply takes far too little literature into account. 

The methodological approach of the analysis is hard to follow. If I understand it correctly, the entire derivation of the conclusions is based on plausibility considerations. The study therefore has more of a scenario-based character. However, it is not described as such. There is also a lack of background information on the assumptions made in order to be able to assess the plausibility of your considerations. I also wonder why this methodological approach was chosen in the first place. In my opinion, a case study with one or more local companies would have been much better suited to assess the applicability and benefits of the circular business model. 

The presentation of the results is detailed, but in my opinion emphasizes the wrong points. The interpretation and further discussion of the results would have been important to me. As described in the introduction, it is not clear to me what is new or unexpected about the results overall and how these findings really advance research or practice. At the end of the article there is a list of points intended to emphasize the scientific and practical added value, but for me this list is too superficial. 

Of secondary importance: right at the beginning of the first page there is one T too many.

Author Response

Dear Reviewer,

Thank you for your valuable feedback on our manuscript. We have carefully considered your comments and made the following revisions to address your concerns:

  1. Introduction and Objective Clarification: We have enriched the introduction with additional references from current scientific research to provide a broader context. We also refined the study's objective to articulate the significance of comparing the financial profitability of linear and circular business models in the wine industry. 

  2. Removal of Technical Description: The technical description at the top of page 3, unrelated to our study, has been removed for clarity and conciseness.

  3. Theoretical Framework and Literature Review: We revised Section 3 to more accurately reflect its content. Additionally, we expanded our literature review to include cross-comparisons with other industries and global practices in waste recycling, providing a more comprehensive understanding of the circular economy's importance.

  4. Methodological Approach: We have revised the methodology section to better elucidate our approach, which involves a mixed-method analysis combining quantitative financial analysis and qualitative assessments. This approach was chosen to provide a holistic view of the circular and linear business models in the Georgian wine industry. We understand your suggestion regarding the use of case studies. However, our approach was chosen to allow for a broader analysis of the industry as a whole, rather than focusing on individual companies. This was deemed more appropriate for our objective of providing a comprehensive overview of the industry-wide transition to circular business models..

  5. Correction of Typographical Error: The typographical error at the beginning of the first page has been corrected.

We believe these revisions have significantly improved the manuscript and appreciate the opportunity to enhance its contribution to the field.

Best regards,

Vakhtang

Round 2

Reviewer 4 Report

Comments and Suggestions for Authors

Dear authors,

Thank you very much for this comprehensive revision, which shows me that you have taken the critical comments seriously. I already like the revised version of your article much better. The presentation of the research objectives and the positioning of the study in the scientific literature have improved significantly. I also noticed positively that the study now refers more strongly to relevant articles in the field.

However, I still have the impression that the literature review is not exhaustive. Important references for your work such as Mura et al. (2023), Cavicchi & Vagnoni (2022) or Sehnem et al. (2020) are not mentioned. The methodological description is also not precise enough for me. You write: "This literature review adopts a systematic approach, methodically evaluating and synthesizing existing research on circular and linear business models in the wine industry. […] By applying rigorous criteria for literature selection and analysis, the review seeks to provide a thorough understanding of the current state of knowledge in this area, highlighting gaps and opportunities for future research." That's fine so far, but what criteria did you use to select the literature? Which databases, which time periods, which search strings? This information is important not only because the literature review is not exhaustive, but also because at many points in the article you indicate that the literature review was thorough and well-organized: " The qualitative component of our research hinges on an exhaustive literature review, encompassing studies on circular and linear economic models both globally and within the Georgian wine industry."

I would also suggest discussing the results of your analysis (e.g. the financial advantages of the circular model over the linear model) in light of the literature. Is this result surprising and unique to the Georgian wine industry or is it consistent with the wine industry in other parts of the world? Perhaps the evidence in general also suggests that the circular model is always superior in financial terms?

Finally, I would like to thank you again for all the changes and the great progress. I think you are on the right way. The paper has already gained in clarity and argumentative strength. In my opinion, however, a further round of revision is necessary.

Wish you all the best.

Author Response

Dear Reviewer,

Thank you once again for your valuable feedback and insightful comments on our manuscript. We acknowledge your concerns regarding the exhaustiveness of our literature review and the precision of our methodological description. 

We have carefully considered your suggestions and have revised our manuscript accordingly. Notably, we have incorporated references to pivotal works such as Mura et al. (2023), Cavicchi & Vagnoni (2022), and Sehnem et al. (2020) within our introduction and literature review sections. These studies, while not primarily focused on the financial analysis and profitability of circular business models, contribute significantly to understanding sustainability issues, CO2 emissions, and their implications for the wine industry. Our intention is to build upon these foundational insights to explore the financial benefits of adopting circular business models, a perspective less covered in existing literature.

Regarding the methodological description, we have now included detailed information about our literature selection process. This includes the databases searched, specific time periods covered, and the search strings used. This enhancement aims to provide clarity on our systematic approach and the rigor behind our literature review.

Concerning the results of our analysis, we appreciate your suggestion to contextualize our findings within the broader wine industry. Our research indicates that the financial advantages of the circular model observed in the Georgian wine industry may not be unique but are consistent with trends observed globally. This suggests that the circular model could potentially offer superior financial benefits across different contexts, a hypothesis that warrants further investigation.

We believe these revisions address your concerns and strengthen our manuscript. We are grateful for the opportunity to refine our work based on your feedback.

Sincerely,

Vakhtang
